# The Loss of Gonadal Hormones Has a Different Impact on Aging Female and Male Mice Submitted to Heart Failure-Inducing Metabolic Hypertensive Stress

**DOI:** 10.3390/cells14120870

**Published:** 2025-06-09

**Authors:** Diwaba Carmel Teou, Emylie-Ann Labbé, Sara-Ève Thibodeau, Élisabeth Walsh-Wilkinson, Audrey Morin-Grandmont, Ann-Sarah Trudeau, Marie Arsenault, Jacques Couet

**Affiliations:** 1Département de Médecine, Faculté de Médecine, Université Laval, Québec City, QC G1V 0A6, Canada; diwaba-carmel.teou.1@ulaval.ca (D.C.T.); emylie-ann.labbe@criucpq.ulaval.ca (E.-A.L.); sara-eve.thibodeau@criucpq.ulaval.ca (S.-È.T.); elisabeth.walsh-wilkinson.1@ulaval.ca (É.W.-W.); audrey.morin-grandmont@criucpq.ulaval.ca (A.M.-G.); ann-sarah.trudeau.1@ulaval.ca (A.-S.T.); marie.arsenault@criucpq.ulaval.ca (M.A.); 2Groupe de Recherche sur les Valvulopathies, Centre de Recherche de l’Institut Universitaire de Cardiologie et de Pneumologie de Québec, Université Laval, Québec City, QC G1V 4G5, Canada

**Keywords:** mouse, heart failure, sex differences, cardiac hypertrophy, aging, preclinical model, HFpEF

## Abstract

Background: Aging and the female sex are considered risk factors for the development of heart failure with preserved ejection fraction (HFpEF). Unlike other risk factors, such as hypertension, obesity, or diabetes, they do not represent therapeutic targets. Methods: In a recently developed two-hit murine HFpEF model (angiotensin II + high-fat diet; MHS), we studied the relative contributions of the biological sex, aging, and gonadal hormones to cardiac remodeling and function. We aimed to reproduce a frequent HFpEF phenotype in mice characterized by aging, hypertension, the female sex, menopause, and metabolic alterations. Using the MHS mouse model, we studied cardiac remodeling and function in C57Bl6/J mice of both sexes, young (12 weeks) and old (20 months), that were gonadectomized (Gx) or not. Results: We observed that in mice, aging was associated with body weight gain, cardiac hypertrophy (CH), left ventricle (LV) concentric remodeling, and left atrial (LA) enlargement. Diastolic parameters such as E and A wave velocities were modulated by aging but only in females. Submitting young and old mice to MHS for 28 days induced the expected HFpEF phenotype consisting of CH, LV wall thickening, LA enlargement, and diastolic dysfunction with a preserved EF except for old males, in which it was significantly reduced. Young mice were Gx at five weeks, and old mice at six months (over a year before MHS). Gx increased myocardial fibrosis in MHS females and helped preserve the EF in males. Conclusions: Our results suggest that MHS has sex-specific effects on old mice, and the loss of gonadal hormones significantly impacts the observed heart failure phenotype.

## 1. Introduction

Aging, the female sex, and menopause are among risk factors for heart failure with preserved ejection fraction (HFpEF) [1,2]. These factors often combine to create a background where hypertension, metabolic alterations such as obesity or type 2 diabetes, atrial fibrillation, kidney disease, valve disease, or other anomalies can trigger the development of HFpEF [3,4]. The relative contribution of aging and menopause in women is difficult to study since these are interrelated.

The prevalence of HFpEF is higher in women than men [5]. Among the proposed reasons to explain this, one is that comorbidities impact women differently [6]. For instance, obesity is more prevalent in women [7], and diabetes is a risk factor more strongly associated with HFpEF [8]. Hypertension is more prevalent in female HFpEF patients [9]. Some factors are sex-specific, such as menopause [10], preeclampsia, and other reproductive factors [5]. All of these and other factors, such as atrial fibrillation and non-obstructive coronary artery disease, may explain the predominance of HFpEF in women [5].

A better understanding of the underlying causes of this sex discrepancy in HFpEF prevalence and outcomes necessitates preclinical models that concentrate not only on traditional risk factors but considers variables such as the biological sex and age.

We developed a “two-hit” murine HFpEF model combining an angiotensin II (AngII) continuous infusion and a high-fat diet (HFD) for 28 days (metabolic-hypertensive stress or MHS) in young male and female C57Bl/6J mice. We also observed that this MHS regimen was efficient in inducing an HFpEF phenotype in old (19 months) ovariectomized (Ovx at the age of 6 months) female four-core genotype (FCG) mice [11]. In this study, we used Ovx as a surrogate for menopause since MHS was only induced a year later [12].

Withaar and collaborators previously developed an HFpEF model in aging mice [13]. They treated old intact females (20 months) with an HFD for four months and added an AngII infusion during the last four weeks. They observed that males evolved towards heart failure with reduced ejection fraction (HFrEF), suggesting that old male mice were not useful preclinical models since this HFpEF transition towards HFrEF is seldom observed in human patients.

Since their model is more severe than ours, we felt that older male and female mice could be used to study the relative impact of aging, the loss of gonadal hormones, and sex differences.

We report that aging is sufficient for the appearance of several cardiac features associated with HFpEF in mice and that Ovx in females can exacerbate these. The MHS in Ovx aging females reproduces the expected HFpEF phenotype, whereas in intact males, several animals evolved towards an HFrEF phenotype. Loss of testosterone by castration completely reversed this and maintained the HFpEF phenotype in old males.

## 2. Materials and Methods

### 2.1. Animals

Four-week-old C57BL6/J mice (male and female) were purchased from Jackson Laboratory (Bar Harbor, ME, USA). Mice were housed on a 12 h light/12 h dark cycle with free access to chow and water. The protocol was approved by the Université Laval’s animal protection committee and followed the recommendations of the Canadian Council on Laboratory Animal Care (#2020-603 and #2020-701). This study was conducted following the ARRIVE guidelines. Mice were randomly distributed into various experimental groups.

Gonadectomy surgical procedures were performed as previously described in young groups at the age of 5 weeks and in old groups at the age of 6 months [11,14]. Old mice had a running device in their cages until the beginning of the experimentation at 19 months [11,12,15].

Regarding metabolic and hypertensive stress (MHS), mice received a continuous infusion of angiotensin II (AngII; 1.5 mg/kg/day) (Sigma, Mississauga, ON, Canada) for 28 days and were fed a high-fat diet (HFD: 60% calories; Research Diets Cat. #D12492) as previously described (Research Diets, New Brunswick, NJ, USA) [11].

For young groups, eight to ten mice of each sex were used. At eight weeks, mice of both sexes were divided into the following groups: controls (Ctrl), ovariectomized (Ocx) or Ovx controls, MHS, and Ocx or Ovx MHS.

For old groups, the experimental design was like that of younger animals, except 19-month-old animals were used.

Experienced technicians monitored the mice’s health and behavior. The animals were weighed weekly. Three intact male mice died in the MHS group.

### 2.2. Echocardiography

Echocardiography was performed under isoflurane anesthesia as described previously [14,16].

### 2.3. Myocardial Fibrosis Evaluation

Myocardial samples were fixed, sliced into serial 10 µm thick sections, and stained with Picrosirius Red to assess the overall percentage of interstitial fibrosis. Interstitial fibrosis was quantified as previously described [11].

### 2.4. Cardiomyocyte Cross-Sectional Area

CSA was measured and expressed in µm^2^ as previously described [17].

### 2.5. RNA Isolation and Quantitative Real-Time Polymerase Chain Reaction

As previously described, quantitative RT-PCR was used to evaluate LV gene expression for at least six animals per group. Cyclophilin A (*Ppia*) was the control “housekeeping” gene. The primers used are listed in Appendix A.

### 2.6. Western Blots

Protein content was estimated by Western blotting as described previously [17]. Antibodies were diluted to a ratio of 1:1000 (PDK4 and p-PDH) or 1:250 (PDH1) in a TSB-T solution with 5% bovine serum albumin (BSA). Phospho-PDH antibody was obtained from Cell Signalling Technologies (Danvers, MA, USA; #31866), whereas PDH (AB110416-1002) and PDK4 antibodies were from Abcam (AB214938) (Toronto, ON, Canada).

### 2.7. Plasma Inflammatory Marker Measurements

The mouse plasma inflammatory protein content was externally characterized by Olink Proteomics^®^ (Uppsala, Sweden) using a proteomics assay simultaneously measuring 43 proteins per sample (Olink Target 48 Mouse Cytokine Panel). In brief, this assay involves protein-specific antibody pairs labeled with unique complementary oligonucleotides (probes) being added to 1 μL of sample in a 96-well plate. Only when both antibodies in the pair bind to the corresponding protein are their attached probes close enough to hybridize. This generates a polymerase chain reaction (PCR) target sequence amplified and detected using a standard real-time PCR protocol. Thirty cytokines of 43 passed the level of quantification and could be measured in the plasma of the animals. Four animals for each of these groups were evaluated: male or female and control or MHS.

### 2.8. Statistical Analysis

All data are expressed as the mean ± standard error of the mean (SEM). Outliers were removed using the ROUT test with a Q of 1% with Prism. Intergroup comparisons were conducted using Student’s *t*-test with GraphPad Prism 10.4 (GraphPad Software Inc., La Jolla, CA, USA). Comparisons of more than two groups were analyzed using a one-way or two-way ANOVA and the Holm–Sidak post-test. *p* < 0.05 was considered statistically significant.

## 3. Results

### 3.1. Short and Long-Term Effects on Cardiac Morphology and Function in Male and Female Mice

As illustrated in Figure 1A, C57Bl6/J mice were either gonadectomized (Gx) at 5 weeks or 6 months, respectively. Younger mice were euthanized 3 months later, and older ones 14 months later. Age-matched sham-operated intact controls were also studied.

Orchiectomy (Ocx) of young and adult males reduced their body weight, and ovariectomy (Ovx) had no effect (Figure 1B). Body growth was also reduced in young Ocx mice (Figure 1C). Gx resulted in a reduced indexed heart weight (iHeart, Figure 1D) except in young males.

We studied these mice by echocardiography just before euthanasia. Using M-mode imaging, we measured the LV internal end-diastolic diameter (EDD; Figure 1E), LV wall end-diastolic thickness (interventricular septal wall thickness + posterior wall thickness; Figure 1F), and the ratio of the two measurements (RWT; Figure 1G) to evaluate LV remodeling caused by both aging and Gx. In males, aging was associated with increased EDD and LV wall thickening and more concentric remodeling, as illustrated by an increased RWT ratio. Ocx strongly reduced EDD and LV wall thickness in older animals, but RWT was maintained. In intact females, EDD was slightly increased by age, but LV walls were thicker, increasing RWT. Ovx reduced LV wall thickening, resulting in less LV concentric remodeling with age. The ejection fraction (EF; Figure 1H) estimated using Simpson’s method remained unchanged in young and old animals, except for old Ocx males displaying a raised EF. Stroke volume (SV; Figure 1I) and cardiac output (CO; Figure 1J) were increased in older intact animals. In Gx animals, SV and CO were reduced compared to intact age-matched controls in old males and young females.

Diastolic function parameters were modulated by age or Gx in these mice. E and A wave velocities (Figure 1K,L) were reduced in Ocx males and increased in Ovx females. Aging increased these velocities in old females but not in young females. The E/A ratio remained unchanged except in young Ocx males (Figure 1M). Tissue Doppler E’ wave velocity was unchanged in all groups (Figure 1N). E/E’ ratios were reduced in Ocx males (Figure 1O) and increased in old females. Finally, Ocx significantly reduced the left atrial tissue weight in young and old males and in old females (Figure 1P). Additional echo results are displayed in Appendix A.

Representative M-mode LV views are illustrated in Figure 2A and LV long-axis sections in Figure 2B, stained using picrosirius red. The quantification of myocardial fibrosis is illustrated in Figure 2C. Aging did not increase this parameter in intact males but did in Ocx ones. In females, aging was associated with a mild increase in myocardial fibrosis. Ocx increased the expression levels of four genes related to extracellular matrix remodeling in older males, namely procollagens 1 and 3 (Col1a and Col3a; Figure 2D,E), periostin (Postn; Figure 2F), and thrombospondin 4a (Thbs4a; Figure 2G). In young males, Col3a and Thbs4 mRNAs were modulated in the opposite direction by Ocx. Aging intact males had lower mRNA levels of these two genes. In females, aging reduced the gene expression of these four genes and Ovx reversed this.

### 3.2. Aging in Female Mice Is Associated with a Circulatory Inflammation Profile

Using Olink^®^ proteomic technology, we measured the levels of 46 inflammatory molecules in the plasma of young and old mice of both sexes, Gx or not. As illustrated in Figure 3, the levels of CCL2, Csf3, IL-17a, IL-17f, IL-6, and Tnf*α* in older females were higher than in their younger counterparts. This was not observed in old males. Aging also reduced the circulatory contents of Cxcl9 and PDCD1LG2, again only in females. The only factor modulated by age in male mice was Ctla4. The rest of the measured inflammation markers were not significantly modulated by age or differed between sexes.

### 3.3. The Effects of Metabolic Hypertensive Stress (MHS) Are Similar in Mice Regardless of Age, Sex, or Gx

Young and old mice, Gx or not, received a continuous AngII infusion for four weeks and were fed a high-fat diet (Figure 4A). As illustrated in Figure 4B,C, body weight (BW) remained stable after MHS in young males and females and old male mice. Old Ovx females had a significant BW reduction after MHS. Heart weight (Figure 4D,E) and left atrial weight (Figure 4F,G) were, as expected, increased by MHS.

We then concentrated on clinically relevant groups: old ovx females, which mimic post-menopausal aging women, and old intact males. Young mice will provide controls for the effects related to aging. Figure 5A,B illustrate that the effects of MHS on cardiac hypertrophy (CH; iHeart) were similar in young and old males. In old ovx females, CH was reduced compared to young controls after MHS. Left atrial enlargement after MHS (iLA; LA weight indexed for tibial length) did not lead to superior values in older animals. However, the baseline LA weight was increased in these mice compared to younger controls (Figure 5C). The levels of heart failure in our animals were mild as lung congestion was not present (Figure 5D).

Echocardiography evidenced that MHS resulted in thickened LV walls and concentric remodeling (Figure 5E,F). This was accompanied by smaller LV internal diameters (EDDs) except for older males (Figure 5G). Several older males also showed significant losses in the LV ejection fraction (Figure 5H). A slight decrease in the EF was observed in older females, but all animals maintained EF values over the normal level (50%). As illustrated in Figure 5I in long-axis LV diastolic and systolic views of representative old male mice (EF near the mean of each group), the MHS in an old male mouse (EF = 40%) resulted in larger LV systolic areas compared to the male control LV tracings.

The cardiomyocyte cross-sectional area (CSA) was increased in older animals compared to young mice (Figure 6A,B). The MHS, as expected, further increased the CSA compared to the controls. Myocardial interstitial fibrosis was more abundant after MHS in young mice (Appendix A). In older animals, the fibrosis levels after MHS were the highest (Figure 6C,D and Appendix A) in males.

**Figure 4 cells-14-00870-f004:**
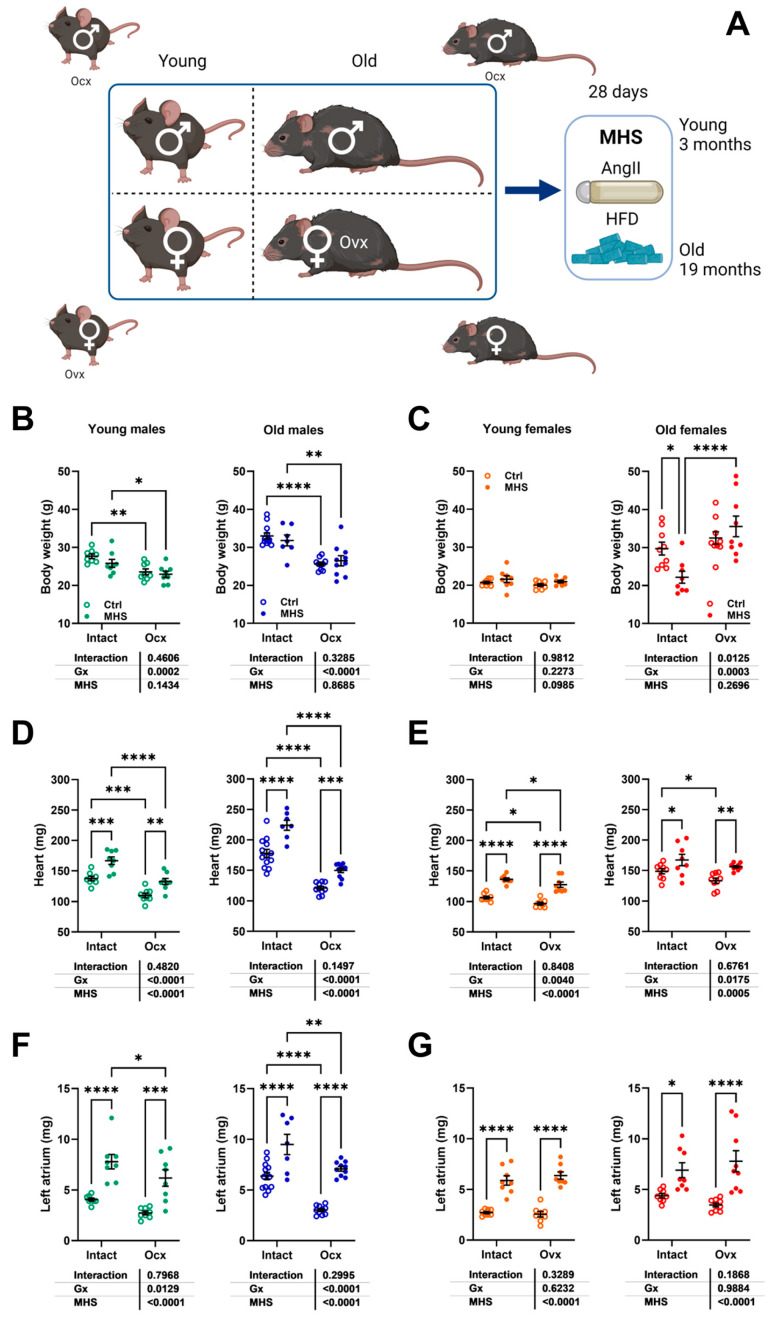
MHS in young and old mice: impact of gonadectomy. (**A**) Schematic representation of experimental design. (Created in BioRender. Couet, J. (2025), https://BioRender.com/aqzg5bj, accessed on 1 May 2025). (**B**) Body weight in young and old intact and Ovx males. (**C**) Body weight in young and old intact and Ovx females. (**D**) Heart weight of males. (**E**) Heart weight of females. (**F**) Left atrial weight of males. (**G**) Left atrial weight of females. Data are presented as mean +SEM (n = 7–14 per group). Two-way ANOVA was performed, followed by Holm–Sidak post-test. * *p* < 0.05, ** *p* < 0.01, *** *p* < 0.001, and **** *p* < 0.0001 between indicated groups.

**Figure 5 cells-14-00870-f005:**
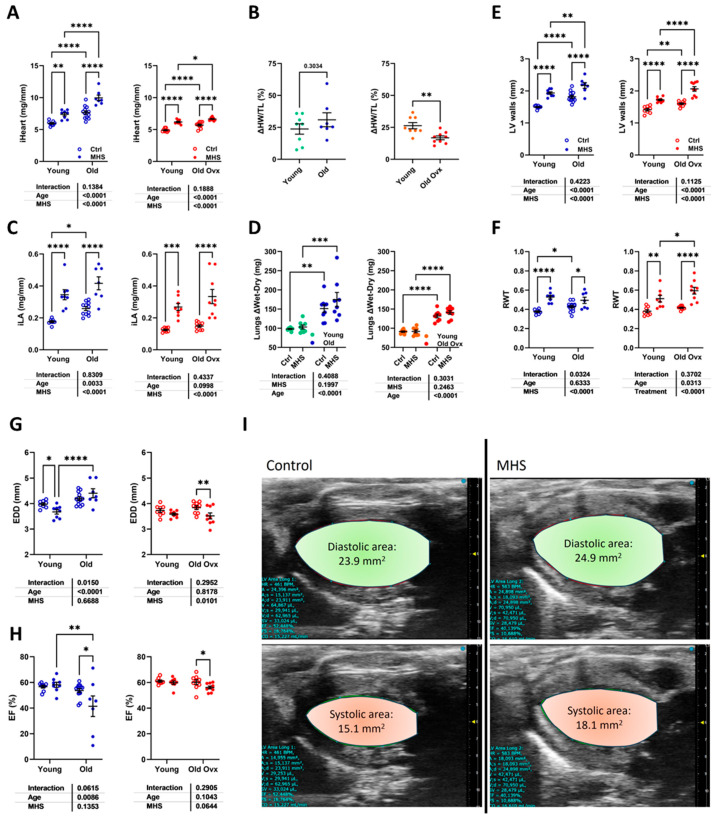
Effects of age (males) and age + Ovx (females) on cardiac response to MHS. (**A**) Indexed heart weight (iHeart; males, left and females, right). (**B**) iHeart gain from MHS in % (HW/TL). (**C**) Indexed left atrial weight (iLA). (**D**) Lung water weight (difference between wet and dry weight). Echocardiography data. (**E**) LV wall thickness in diastole (PWd + IVSd). (**F**) LV relative wall thickness (LV walls/EDD; RWT). (**G**) End-diastolic LV diameter (EDD) and (**H**) ejection fraction (EF). Data are presented as mean ± SEM (n = 7–14 per group). Two-way ANOVA was conducted, followed by Holm–Sidak post-test. * *p* < 0.05, ** *p* < 0.01, *** *p* < 0.001, and **** *p* < 0.0001 between indicated groups. (**I**) B-mode LV diastolic tracings of an old control male (left; EF: 52.3%) and an old MHS male (right; EF: 40.1%).

**Figure 6 cells-14-00870-f006:**
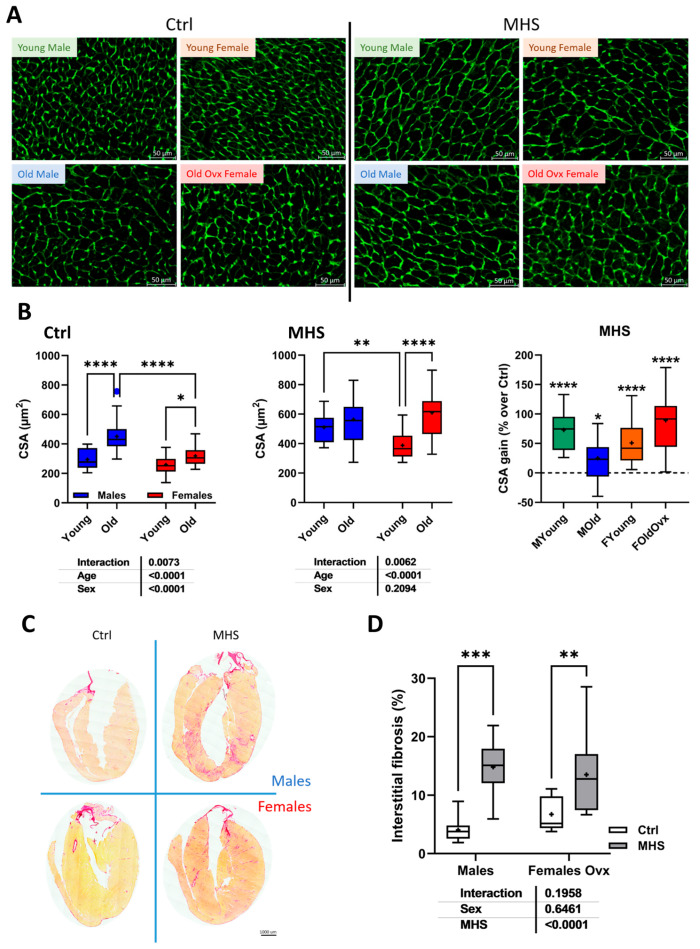
Cardiac myocyte hypertrophy and myocardial interstitial fibrosis after MHS. (**A**) Representative images of WGA-FITC staining from LV sections of various indicated groups. (**B**) Cross-sectional area of cardiomyocytes quantified by WGA-FITC staining control (Ctrl; left) and MHS (middle) groups. CSA gain (% over control; right) from MHS. (**C**) Representative images of picrosirius red staining of old female and male heart sections (Ctrl vs. MHS). (**D**) Myocardial fibrosis (picrosirius red staining). Data are presented as mean ± SEM (n = 7–8 per group). Two-way ANOVA was conducted, followed by Holm–Sidak post-test. * *p* < 0.05, ** *p* < 0.01, *** *p* < 0.001, and **** *p* < 0.0001 between indicated groups.

### 3.4. In Young Female Mice, MHS Induced a Circulatory Inflammation Profile Similar to Aging

We measured the levels of 46 inflammatory molecules in the plasma of young mice of both sexes in the control and MHS groups. As illustrated in Appendix A, the levels of Il-6, Csf3, Il-17a, Il-10, Il-16, Ccl11, and Pdcdlg2 were significantly higher in MHS females. In males, only Il17f showed higher plasma concentrations than the controls.

### 3.5. Modulation of Myocardial Hypertrophy and Fibrosis Marker Genes by MHS

The expression levels of several myocardial hypertrophy or fibrosis marker genes were measured to highlight differences in modulation related to aging or sex. As illustrated in Figure 7A,B, MHS increased the LV gene expression of natriuretic peptides, Nppa (atrial), and Nppb (brain). For Nppa, this increase was similar irrespective of the age or sex of the animals. Nppb mRNA levels were more strongly increased in young males than in old mice.

Procollagens 1 and 3 genes (Col1a1 and Col3a1) were increased in young and old males after MHS (Figure 7C,D). In females, MHS did not significantly increase the expression levels of these two genes in young animals, but only in older ones. Periostin (Postn) and thrombospondin 4 (Thbs4), marker genes of extracellular matrix remodeling, had increased expression levels after MHS in young and old males. In contrast, they were modulated more markedly in old females (Figure 7E,F).

### 3.6. MHS Increases the Pyruvate Dehydrogenase Kinase 4 (PDK4) Protein Content in Young and Old HFpEF Male and Female Mice

We recently observed that MHS possibly blocked the use of glucose as a myocardial energy substrate by reducing glucose entry into the cell via glucose transporter 4 (Glut4) and pyruvate entry into the mitochondrion via pyruvate dehydrogenase (PDH) phosphorylation by pyruvate dehydrogenase kinase 4 (Pdk4) [11]. We were interested to see whether this shift in myocardial energy substrate preference was present in older animals.

We measured the mRNA levels of genes encoding for fatty acid (CD36/FAT) entry into the cell and one in their entry into the mitochondrion (carnitine palmitoyl dehydrogenase 1B, Cpt1b). As illustrated in Figure 8A,B, those two genes are less expressed in older animals and not controlled by MHS except for younger males (CD36/FAT). Glut4 gene expression was also reduced in the myocardium of older animals and was significantly reduced by MHS in females (Figure 8C). Pdk4 mRNA levels increased after MHS in young and old mice (Figure 8D).

BDH1 (3-Hydroxybutyrate Dehydrogenase 1) is involved in ketone body metabolism. Age increased Bdh1 gene expression in males and had the opposite effect in females (Figure 8E). MHS increased its expression in males, while a similar tendency (*p* = 0.08) was observed in females. As for the AMP kinase subunit B1 (Ampkb1) gene expression, age reduced the mRNA levels in males, and MHS had the same effect in females, but not with age (Figure 8F).

### 3.7. Loss of Testosterone Rescues LV Systolic Function in Old MHS Males

As illustrated in Figure 5E, MHS in old male mice led to three premature deaths and EF levels below 50% in five of the seven surviving animals. Figure 9A shows that gonadectomy reduced the indexed heart weight after MHS in males but not females. Ejection fraction loss after MHS was also completely reversed in old males, and LV dilation was prevented (Figure 9B). Ocx did not change Nppa, Col1a, Col3a, and Postn mRNA levels in old MHS males but increased Nppb and Thbs4 expression (Figure 9C,D). Thbs4 levels were more enhanced in Ovx females compared to intact old ones.

## 4. Discussion

We previously described a HFpEF two-hit mouse model combining AngII continuous infusion and an HFD for 28 days [11]. In this study, we studied factors such as aging and gonadectomy that could influence cardiac remodeling and function in mice of both sexes. We used young animals as controls to highlight changes occurring with aging. In parallel, we studied mice of both sexes using gonadectomy as a surrogate of menopause in aging females and to potentially explain observed differences related to biological sex. A diagram summarizing our findings is illustrated in Figure 10.

In mice of both sexes, aging was associated with cardiac hypertrophy, LV concentric remodeling, and left atrial enlargement. The ejection fraction remained stable. Diastolic function was altered only in females. We let our aging mice exercise voluntarily (VE) for 6 to 20 months, which could have helped them maintain better cardiac health. Diastolic dysfunction has previously been reported in sedentary old mice. In a recent study in old mice (24 months), we observed that they maintained relatively stable diastolic and systolic functions when allowed to VE, reducing unwanted and destructive behaviors (aggression, barbering, etc.) and better reproducing their natural life in the wild [12,15]. These findings align with previous observations showing cardiac hypertrophy and LA enlargement in aging male mice [18,19,20]. However, we did not observe alterations in the diastolic echocardiographic parameters we measured. A primary difference in our study is that we let our mice exercise during aging and up to MHS. Due to this, our old mice may have had better cardiac and overall health than the sedentary mice. Exercise has been previously shown to reverse several phenotypes associated with cardiac aging, including diastolic dysfunction in aging male mice [21]. Our study did not include old sedentary mice; it is thus difficult for us to pinpoint the effects of VE on aging. Unlike the studies mentioned above, diastolic function in our old mice thus remained mostly normal, myocardial fibrosis only slightly increased, and the myocardial expression of marker genes remained unchanged or lower than that in young mice.

The loss of gonadal hormones at 6 months and 14 months reduced cardiac hypertrophy related to aging in male and female mice. Old Gx males were leaner than Ovx females. Gx at 6 months did not inhibit body growth. LV wall thickening decreased in gonadectomized animals. Moreover, in males, the LV diameter was smaller, which resulted in smaller stroke volume and cardiac output. Diastolic function in older Gx females was similar to that of intact ones.

We observed in male mice that aging changed the cardiac phenotype achieved after MHS from the expected HFpEF to HFrEF as previously observed by others in a similar model [13]. Moreover, three males died during MHS, suggesting that these, too, could have evolved towards HFrEF. The loss of gonadal steroids completely reversed the EF decrease and LV dilation. Our results suggest that more work is needed to develop a more clinically relevant HFpEF mouse model in older males. Our initial assumption was that shortening the time that the high-fat diet was administered to mice from four months to one month [13] could prevent the evolution towards HFrEF. This was not the case with less severe stress either.

Few studies have been conducted on the effects of long-term deprivation of gonadal steroids on the heart in mice. In females ovariectomized at 1 month and then studied at 24 months, compared to sham-operated animals, intracellular Ca^2+^ dysregulation, reduced myofilament Ca^2+^ sensitivity, and increased spontaneous Ca^2+^ release were observed. Structural cardiac changes were not reported [22]. A similar experimental design was reproduced in male C57B6/J mice, except the animals were studied at 18 months. Gonadectomy was associated with smaller hearts, longer IVRT, and lower E/A ratios, suggesting diastolic dysfunction developing with age [23]. Our mice were gonadectomized later in life at 6 months and were allowed to exercise. We observed that gonadectomy can influence cardiac function both during a young age (males) and later in life for both sexes. We previously reported that cardiac growth in males happens mostly during their first year of life, slowing down afterwards [12]. By performing gonadectomy at 6 months, we decreased normal cardiac growth in old males to a certain extent, but it probably would have decreased more if it had been performed sooner in life. In females, cardiac growth is more gradual during the first two years of life. We confirmed our previous observations that gonadal steroids had little influence on cardiac growth in young females, but their absence slowed this growth (or hypertrophy) later in life [12,15].

Gonadectomy in older mice was associated with an increased myocardial fibrosis content and higher gene expression levels of *Col1*, *Col3*, *Postn*, and *Thbs4*. In females, the mRNA levels for these four genes were also increased in old Ovx mice. This was not observed in young animals. It is unclear why the loss of gonadal steroids, more than a year after Gx, was still associated with the activation of extracellular matrix (ECM)-related genes. Moreover, *Col1* (females), *Col3*, and *Thbs4* gene expression were reduced in older intact mice compared to young controls. It is possible that myocardial remodeling during cardiac growth in young mice requires higher levels of ECM-related gene expression, which is not the case for older animals.

We used Gx as a surrogate for menopause. We chose to study the animals late after the surgery to concentrate on the long-term cardiac effects of aging in a “post-menopausal” state. Two recent studies in HFpEF mice used a chemical method (vinylcyclohexene dioxide or VCD) to induce ovarian failure (menopause) in mice [24,25]. One was conducted in the C57Bl6/J strain, as in our study, and the other in the C57Bl6/N strain [24]. Both used the L-NAME + HFD two-hit HFpEF murine model in young animals. These two inbred mouse strains show differences in the vulnerability of females to develop HFpEF under this regimen, with the J strain being more susceptible. Troy and collaborators reported that ovary-intact menopause did not exacerbate HFpEF development, whereas Methawasin and collaborators observed the opposite [25]. Differences in the mouse strains and experimental designs have been reported to explain this discrepancy [19]. In one study, HFpEF-causing stress was initiated during VCD treatment, while in the other, it was initiated after the perimenopausal period. Both studies were conducted in young mice to exclude aging as a confounding factor.

Interestingly, older female mice had a sex-specific plasma inflammation profile characterized by a marked increase in interleukin-17 (A and F), suggesting an involvement of T helper 17 (Th17) cells during aging. This inflammatory profile was not observed in males. This increase in IL-17 was also present in young MHS female mice. The meaning of this sex difference is unclear and will require confirmation and additional research. It is unclear whether these changes in IL-17 plasma levels affect the heart or indicate that aging and the MHS in female mice trigger a similar response from IL-17-producing T cells. Myocardial infiltration by T cells has been reported in various murine heart disease models, but there are fewer studies in the literature about Th17 cells, the leading producers of IL-17 [26,27,28,29].

Links between IL-17 and heart failure have been observed before, however [30,31,32,33]. In heart failure patients, plasma IL-17 levels were higher than in those without heart failure. These IL-17 levels were negatively correlated with the ejection fraction, suggesting that HFrEF was probably linked to this rise. In male mice with transverse aortic constriction (a LV pressure overload model), IL-17a circulatory and myocardial levels were increased, linking this interleukin to the development of heart failure. In addition to raised IL-17 levels, we observed that IL-6 (interleukin-6) and TNFα (tumor necrosis factor alpha) levels were increased in older females [34,35].

MHS in young mice produced similar cardiac effects irrespective of the presence or absence of gonadal steroids. The main differences were that young Ocx males were smaller and had a smaller heart at baseline. The influence of Ocx on cardiac growth in older males was also apparent, highlighting the role of androgens [36,37].

In females, Ovx did not influence cardiac growth in young animals and cardiac hypertrophy later in life, as we previously reported [12].

MHS led older females towards LV concentric remodeling with a maintained ejection fraction. In contrast, this stress in old males was associated with a significant loss of EF, including 3 animals out of 10 that did not survive the 28-day stress period. This observation is similar to the one previously reported by Withaar et al. [13] in their HFpEF aging model; they also observed that older males developed HFrEF instead of HFpEF. In this HFpEF model, these authors also showed that the two classes of anti-diabetic drugs (glucagon-like peptide receptor agonist (GLP-1 RA) and sodium-glucose co-transporter 2 inhibitors) were recently demonstrated to provide benefits in HFpEF patients, the same as what was observed in old female mice. More recently, they also observed that the GLP-1 RA, semaglutide, induced favorable cardiometabolic effects not directly associated with weight loss in the same mouse model [38].

Here, we show that Ocx can prevent this and return the phenotype towards HFpEF, suggesting that androgens are a critical factor in the heart’s response to this type of stress. As with other cardiovascular disease animal models, sexual dimorphisms are often apparent when a component of pressure overload is present [36]. Male animals consistently develop more severe disease symptoms in various experimental settings. Male hearts are more prone to develop eccentric cardiac hypertrophy, whereas female hearts exhibit concentric hypertrophy. This is also true in patients with aortic stenosis [39]. This eccentric hypertrophy pattern will accompany more myocardial fibrosis and decreased contractility. In our model, fibrosis was not more important in old MHS males than in old Ovx females, but contractility was severely affected. The expression of cardiac marker genes such as *Nppa*, *Nppb*, or collagens was not more modulated in old males than in females.

The effects of androgens on the aging heart are still not completely understood. In aging men, the decrease in androgen levels follows a gradual slope that is probably not different from that of male mice. Testosterone deficiency has been associated with increased mortality in multiple cohort studies; it remains unclear whether this is a causal relationship. Low testosterone levels may be a marker of poor health in aging and the associated disease burden. Current evidence suggests that physiologic androgen levels are beneficial to male cardiovascular health and that deficiency is associated with unfavorable outcomes [36,37]. Reconciliating our observations in old male mice with the clinical setting is hazardous. An extended period of hypogonadism was imposed on young mice, and it seemed to rescue their cardiac health after MHS. This loss of testosterone may push the heart towards diastolic dysfunction instead of lower contractility. Older males have increased myocardial fibrosis compared to age-matched controls and a lower E/E’ ratio, suggesting a state more prone to evolve towards HFpEF when stressed.

Observations in the L-NAME + HFD mouse model indicate that glucose utilization is partially reduced as a substrate for myocardial ATP production via the downregulation of PDH activity [40,41]. We observed that MHS strongly increased the myocardial PDK4 content, but this translated into increased levels of PDH phosphorylation only in young animals and old females. This confirms the observations we made previously in this model using young mice [17].

In the MHS model, because of the lipolytic action of AngII [42], young mice did not gain weight after being fed a high-fat diet for 28 days. Young intact females even lost weight after MHS, but not Ovx mice. The loss of estrogens is usually associated with body weight gain in mice [11], but here, this weight gain was prevented due to AngII. Older mice were more obese than young animals, as expected. Still, voluntary exercise likely reduced the extent of obesity development, which could have contributed to a milder response to MHS.

### Study Limitations

Affirming that MHS similarly impacted young male and female mice is difficult. The cardiac response was similar at the morphological and functional levels, but possible hidden factors must be identified to confirm this.

We did not monitor the mice’s running activity since they were not housed individually. Males are expected to run less than females, and the distance covered diminishes with age. In the wild, mice cover significant daily distances to ensure their subsistence. We consider that providing a running device would help to improve their environment by allowing natural murine behavior.

We did not include hormone replacement groups in our study. Since old Gx animals were kept for over a year, hormone replacement therapy (HRT), although possible, would have been problematic to administer. In addition, it is challenging to design HRT regimens that mimic the real situation in aging animals.

This study was conducted using the C57Bl6/J mouse strain, as are most studies in the field. Studies conducted using inbred strains potentially have a more limited translational value. It will be essential to confirm the validity of these results in other strains or in outbred mice with MHS or other HFpEF-inducing models [43,44].

## 5. Conclusions

In conclusion, aging exacerbated the cardiac response to MHS in old mice. We observed that old male mice cannot be used as HFpEF preclinical models since they are more prone to decompensate towards eccentric LV hypertrophy and a loss of contractility. Old ovx females, on the other hand, do not represent a worsened phenotype compared to intact females at the morphological or functional level. For them, aging is associated with increased LV gene expression levels of four ECM-related genes.

Gonadectomy of old male mice slowed the development of cardiac hypertrophy related to aging and helped the myocardium withstand MHS. This finding is challenging to translate to humans, but it emphasizes the need for further research on the cardiac effects of gonadal hormones in the context of aging.

## Figures and Tables

**Figure 1 cells-14-00870-f001:**
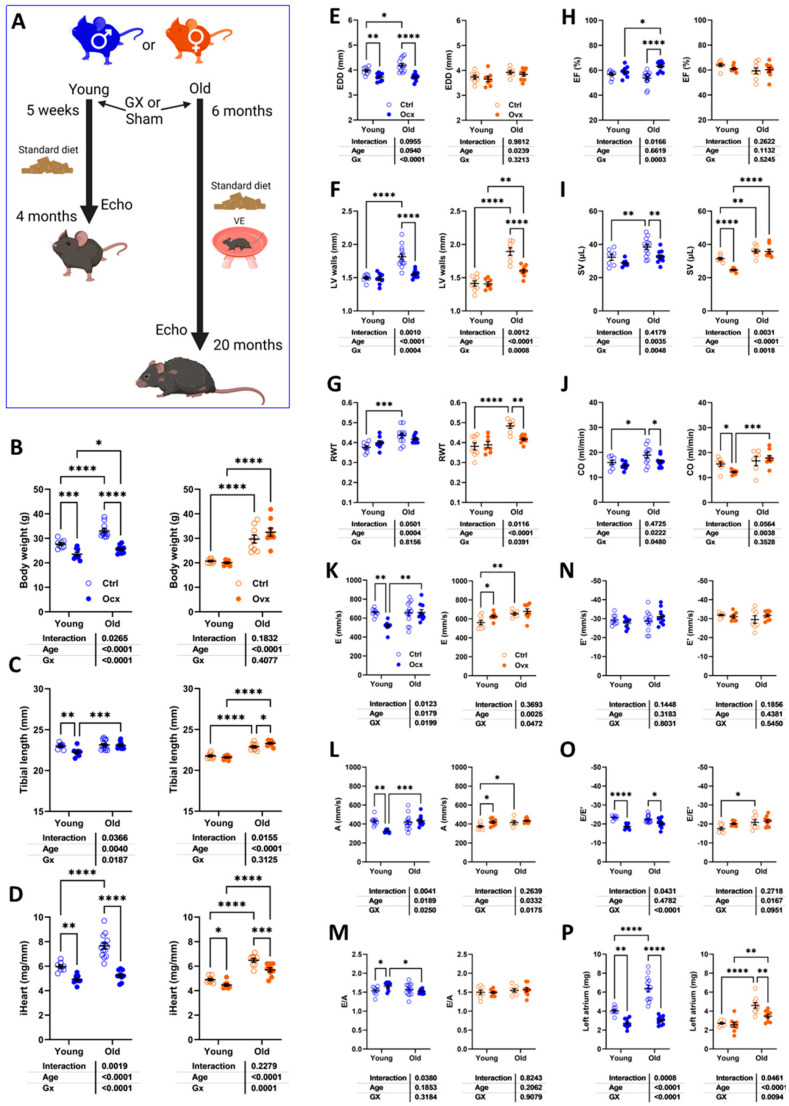
The effects of gonadectomy (Gx) on cardiac morphology and function in young (4 months) and old (20 months) mice of both sexes. (**A**) A schematic representation of the experimental design (created in BioRender. Couet, J. (2025), https://BioRender.com/4zt874x, accessed on 1 May 2025. Male and female C57Bl6/J/J mice were Gx at 5 weeks or 6 months and were allowed to age until 4 and 20 months. All mice were fed a standard diet as described in Section 2. Older animals had a running device installed in their cages after Gx to prevent unwanted behavior and for environmental enrichment. (**B**) The body weight, (**C**) tibial length, and (**D**) indexed heart weight (iHeart) of male (blue) and female (orange) mice. Gx animals (Ocx or ovx) are represented as solid dots. Echocardiography data. (**E**) End-diastolic LV diameter (EDD). (**F**) LV wall thickness in diastole (PWd + IVSd). (**G**) LV relative wall thickness (LV walls/EDD; RWT), (**H**) The ejection fraction (EF). (**I**) LV stroke volume (SV), (**J**) Cardiac output (CO). Diastolic function. (**K**) Pulsed-wave Doppler E wave velocity (E wave). (**L**) A wave velocity and (**M**) E/A ratio. (**N**) Tissue Doppler E’ wave velocity (E’ wave), (**O**) E/E’ ratio, and (**P**) left atrial weight. The results are expressed as the mean ± standard error of the mean (SEM). A two-way ANOVA was conducted, followed by the Holm–Sidak post-test. * *p* < 0.05, ** *p* < 0.01, *** *p* < 0.001, and **** *p* < 0.0001 between indicated groups (n = 8–14 mice/group).

**Figure 2 cells-14-00870-f002:**
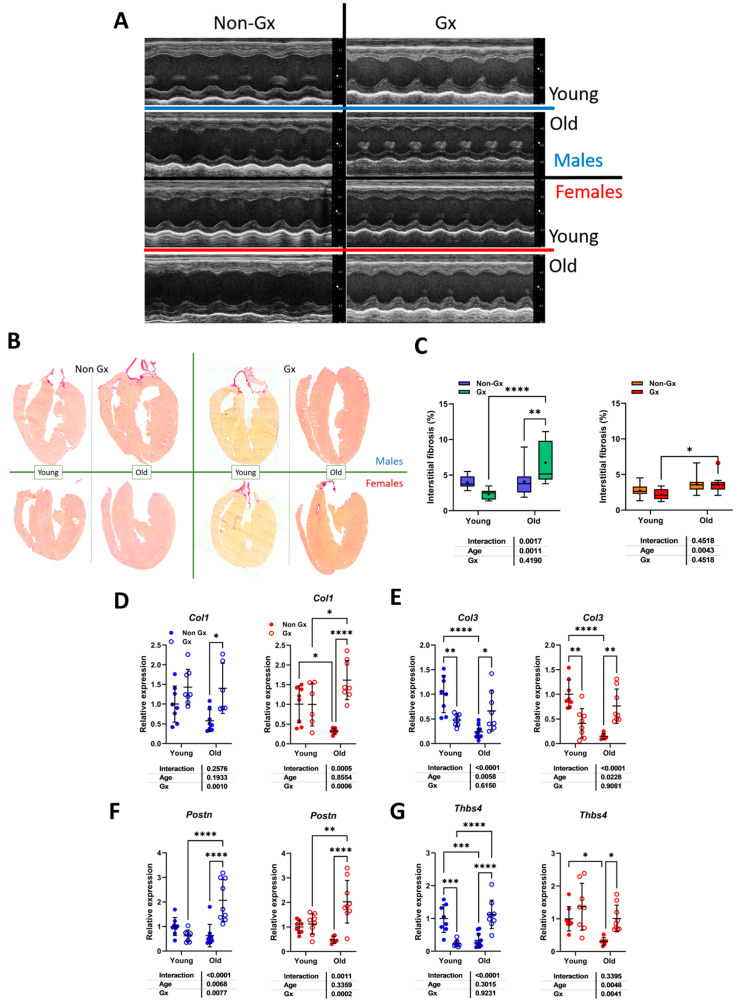
Effects of aging on cardiac morphology and development of myocardial interstitial fibrosis. (**A**) Representative M-mode echo LV tracings of young and old mice, Gx or not. (**B**) Representative images of picrosirius red staining of male and female heart long-axis sections for each indicated group. (**C**) Myocardial fibrosis (picrosirius red staining) in males (**left**) and females (**right**). (**E**) Left graph represents females (red), and right graph represents males (blue). (**D**) *Col1a1*, Collagen 1 α1; (**E**) *Col3a1*, Collagen 3 α1; (**F**) *Postn*, periostin; and (**G**) *Tbsp4*, thrombospondin 4. Data are represented as mean ± SEM (n = 6–8 per group). Two-way ANOVA was conducted, followed by Holm–Sidak post-test. * *p* < 0.05, ** *p* < 0.01, *** *p* < 0.001, and **** *p* < 0.0001 between indicated groups.

**Figure 3 cells-14-00870-f003:**
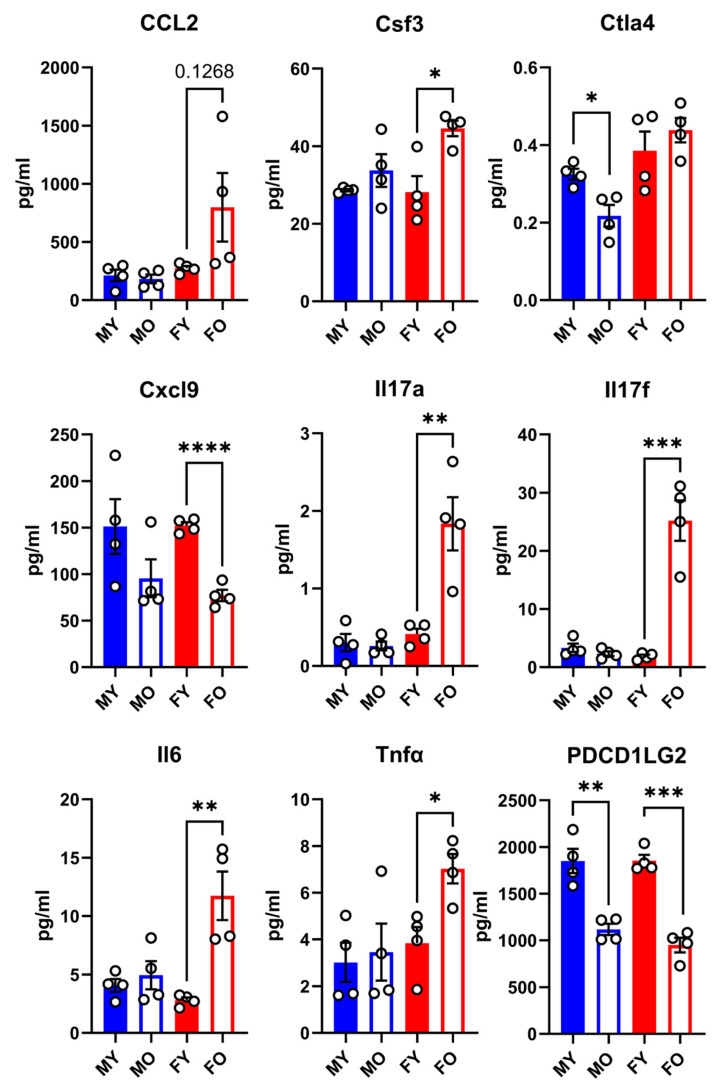
Plasma protein concentration of 9 immunity-related molecules in male and female mice exposed or not exposed to MHS. Levels of these molecules were evaluated as described in Section 2. CCL2: C-C Motif Chemokine Ligand 2, Csf3: Colony-Stimulating Factor 3, Ctla4: Cytotoxic T-lymphocyte associated protein 4, Cxcl9: C-X-C Motif Chemokine Ligand 9, IL-17: interleukin 17, Il6: interleukin-6, Tnfα: tumor necrosis factor alpha, Pdcd1lg2: Programmed Cell Death 1 Ligand 2. Data are represented as mean ± SEM (n = 4). F: females; M: males. Student’s *t*-test was performed for young and old animals. * *p* < 0.05, ** *p* < 0.01, *** *p* < 0.001, and **** *p* < 0.0001 between indicated groups.

**Figure 7 cells-14-00870-f007:**
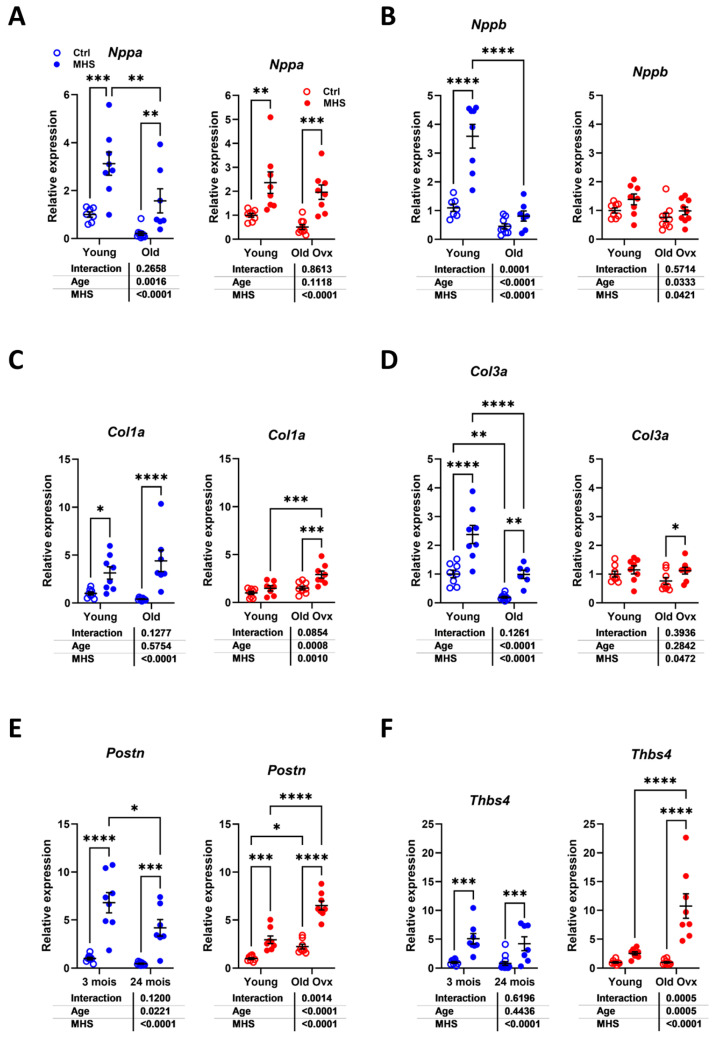
Modulation of LV gene expression after MHS in young and old mice. (**A**) Nppa, atrial natriuretic peptide. (**B**) Nppb, brain natriuretic peptide. (**C**) Col1a1, Collagen 1 *α*1; (**D**) Col3a1, Collagen 3 *α*1; (**E**) Postn, periostin; and (**F**) Tbsp4, thrombospondin 4. Data are presented as mean ± SEM (n = 7–8 per group). Two-way ANOVA was conducted, followed by Holm–Sidak post-test. * *p* < 0.05, ** *p* < 0.01, *** *p* < 0.001, and **** *p* < 0.0001 between indicated groups.

**Figure 8 cells-14-00870-f008:**
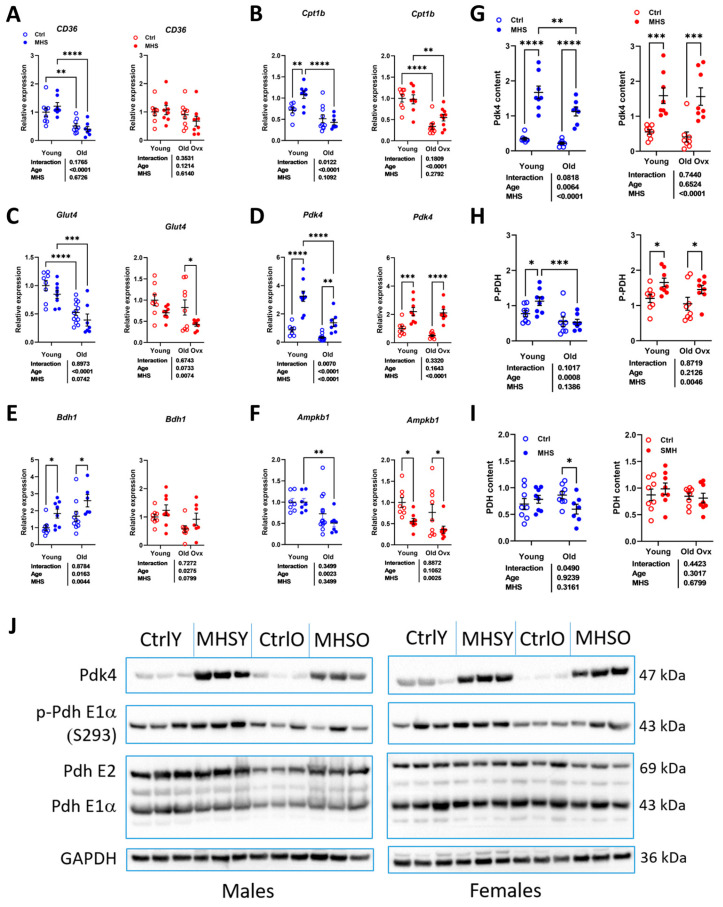
MHS inhibits myocardial energy metabolism via inhibition of glucose utilization. A–D. Expression levels of various genes implicated in myocardial energy production. (**A**) CD36/FAT, fatty acid transporter; (**B**) Cpt1b, carnitine palmitoyl transferase 1b; (**C**) Glut4, glucose transporter 4; (**D**) Pdk4, pyruvate dehydrogenase kinase 4; (**E**) Bdh1, 3-Hydroxybutyrate Dehydrogenase 1; and (**F**). Ampkb1, AMP kinase b1. Protein content estimated by immunoblotting. (**G**) PDK4, (**H**) phospho-PDH (pyruvate dehydrogenase), and (**I**) PDH. (**J**) Representative blots of PDK4, p-PDH, PDH, and GAPDH (Glyceraldehyde 3-phosphate dehydrogenase) contents in young controls (CtrlY), old controls (CtrlO), young MHS mice (MHSY), and old MHS (MSHO) mice. Two-way ANOVA was conducted, followed by Holm–Sidak post-test. * *p* < 0.05, ** *p* < 0.01, *** *p* < 0.001, and **** *p* < 0.0001 between indicated groups.

**Figure 9 cells-14-00870-f009:**
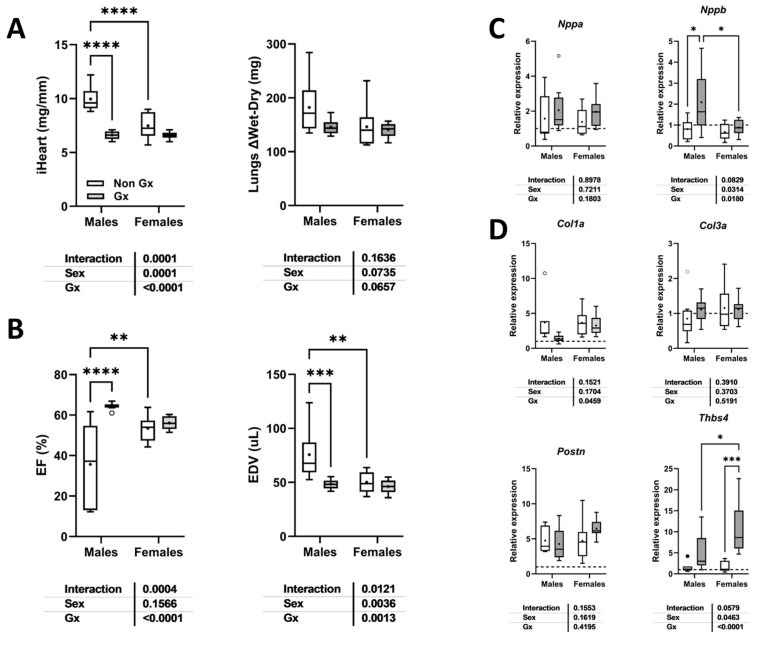
Loss of gonadal hormones modulates cardiac response to MHS in old male mice. (**A**) Indexed heart weight and lung water weight (difference between wet and dry weights); (**B**) ejection fraction and end-diastolic LV volume (EDV). (**C**) Nppa and Nppb LV mRNA levels. (**D**) Col1a, Col3a, Postn, and Thbs4 LV mRNA levels. Two-way ANOVA was conducted, followed by Holm–Sidak post-test. * *p* < 0.05, ** *p* < 0.01, *** *p* < 0.001, and **** *p* < 0.0001 between indicated groups.

**Figure 10 cells-14-00870-f010:**
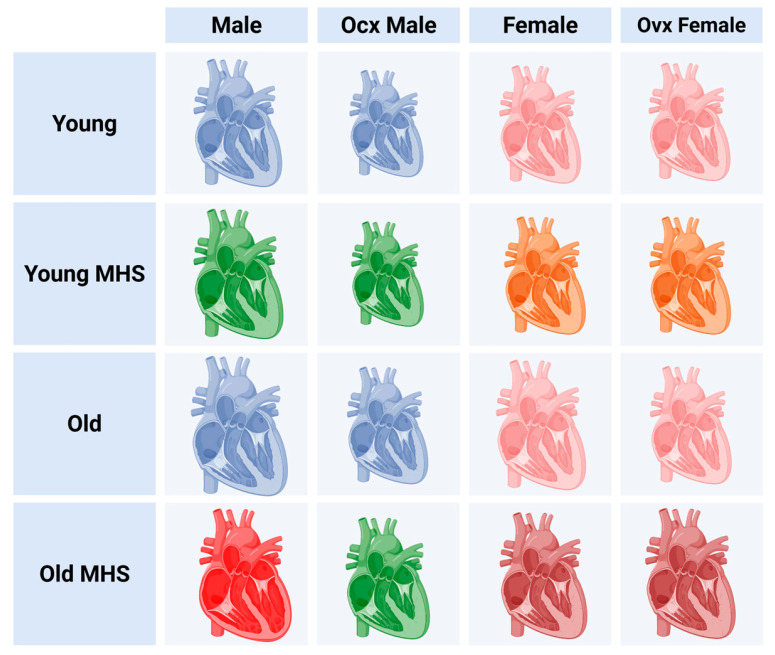
Summary of observations made in this study. In young animals, Gx reduced cardiac growth more in males than in females, and hearts of Ocx males were smaller after MHS or in old animals. Gx in females reduced cardiac growth (hypertrophy) in old animals. MHS resulted in all groups in HFpEF-like phenotype characterized by cardiac hypertrophy, LV concentric remodeling, and left atrial enlargement except in old ocx males, which evolved towards LV eccentric remodeling and lower ejection fraction (HFrEF). Created in BioRender. Couet, J. (2025) https://BioRender.com/g0dpcwr, accessed on 1 May 2025.

## Data Availability

The authors will make the raw data supporting this article’s conclusions available upon request.

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
