# Peer review of "The Loss of Gonadal Hormones Has a Different Impact on Aging Female and Male Mice Submitted to Heart Failure-Inducing Metabolic Hypertensive Stress"

_cells, 2025, doi:10.3390/cells14120870_

Round 1
Reviewer 1 Report
Comments and Suggestions for Authors
The study is an excellent contribution to the study of heart failure using preclinical models. The quality of the experimental design and presentation of the experimental data is very high. The authors hint towards possible mechanisms that would explain their observations. Further studies on these mechanisms, for instance their contribution to fibrosis, would certianly have an impact in our understanding of heart failure and specifically of the forms with preserved ejection fraction. Conclusions are clearly justified by the data and clearly presented.
Author Response
Our comments in italics.
The study is an excellent contribution to the study of heart failure using preclinical models. The quality of the experimental design and presentation of the experimental data is very high. The authors hint towards possible mechanisms that would explain their observations. Further studies on these mechanisms, for instance their contribution to fibrosis, would certianly have an impact in our understanding of heart failure and specifically of the forms with preserved ejection fraction. Conclusions are clearly justified by the data and clearly presented.
Thank your for your kind comments.
Reviewer 2 Report
Comments and Suggestions for Authors
This paper characterizes a mouse model of EFpEF. Experiments are well performed, data are well presented, and the manuscript is well written.
The Introduction does not have much background. It is essential that the authors provide more detailed information about the effects of aging, female sex, and menopause on HEpEF in humans.
In the Abstract, authors state “ We observed that aging was associated in mice with body weight gain, cardiac hypertrophy (CH), left ventricle (LV) concentric remodelling, left atrial (LA) enlargement and changes in echocardiography diastolic parameters (E and A wave velocities), but only in females.” However, authors state in the Discussion section that “In mice of both sexes, aging was associated with cardiac hypertrophy, LV concentric remodelling, and left atrial enlargement.” — Please clarify this inconsistency.
No new mechanistic information has been provided in this study, at least, not clear from the manuscript.
Author Response
This paper characterizes a mouse model of EFpEF. Experiments are well performed, data are well presented, and the manuscript is well written.
The Introduction does not have much background. It is essential that the authors provide more detailed information about the effects of aging, female sex, and menopause on HEpEF in humans.
We added some background related to this in the introduction.
In the Abstract, authors state “ We observed that aging was associated in mice with body weight gain, cardiac hypertrophy (CH), left ventricle (LV) concentric remodelling, left atrial (LA) enlargement and changes in echocardiography diastolic parameters (E and A wave velocities), but only in females.” However, authors state in the Discussion section that “In mice of both sexes, aging was associated with cardiac hypertrophy, LV concentric remodelling, and left atrial enlargement.” — Please clarify this inconsistency.
We thank the reviewer for pointing out this inconsistency. In fact, echo diastolic parameters were only modulated in females by aging. We made the correction in the abstract.
No new mechanistic information has been provided in this study, at least, not clear from the manuscript.
It is true that this manuscript is more descriptive than mechanistic. We are still in the process of fine-tuning our HFpEF model and finding the most clinically relevant way to take into account age and menopause. We reworked the discussion section to highlight pass work from other authors studying loss of sex steroids and aging in the context of CV diseases.
Reviewer 3 Report
Comments and Suggestions for Authors
This is a study investigating about the impact of biological sex, aging and gonadal hormones to cardiac remodelling and function in HFpEF mice model. They studied cardiac remodelling and function in C57Bl6/J mice of both sexes, young (12 weeks) and old (20 months), gonadectomized (Gx) or not. Gx increased myocardial fibrosis in MHS females and helped preserve EF in males, suggesting that MHS has sex-specific effects in old mice, and loss of gonadal hormones significantly impacts the heart failure observed phenotype.
The study was finely arranged.
There were only minor comments for the article.
I think it would be helpful for readers to understand if the findings from this study were presented in a simple diagram.
Author Response
This is a study investigating about the impact of biological sex, aging and gonadal hormones to cardiac remodelling and function in HFpEF mice model. They studied cardiac remodelling and function in C57Bl6/J mice of both sexes, young (12 weeks) and old (20 months), gonadectomized (Gx) or not. Gx increased myocardial fibrosis in MHS females and helped preserve EF in males, suggesting that MHS has sex-specific effects in old mice, and loss of gonadal hormones significantly impacts the heart failure observed phenotype.
The study was finely arranged.
There were only minor comments for the article.
I think it would be helpful for readers to understand if the findings from this study were presented in a simple diagram.
We added a figure summarizing our findings.
Round 2
Reviewer 2 Report
Comments and Suggestions for Authors
In the Abstract, authors state “ We observed that aging was associated in mice with body weight gain, cardiac hypertrophy (CH), left ventricle (LV) concentric remodelling, left atrial (LA) enlargement and changes in echocardiography diastolic parameters (E and A wave velocities), but only in females.” However, authors state in the Discussion section that “In mice of both sexes, aging was associated with cardiac hypertrophy, LV concentric remodelling, and left atrial enlargement.” — Please clarify this inconsistency.
We thank the reviewer for pointing out this inconsistency. In fact, echo diastolic parameters were only modulated in females by aging. We made the correction in the abstract.
Shouldn't the authors make corrections in the Discussion on the statement "In mice of both sexes, ...."?
Author Response
Shouldn't the authors make corrections in the Discussion on the statement "In mice of both sexes, ...."?
We made the correction in the abstract only to resolve the inconsistency. Aging was associated in both sexes with body weight gain, cardiac hypertrophy, left ventricle concentric remodelling and left atrial enlargement. Echo diastolic parameters were changed only in females.
In the discussion, we wrote: In mice of both sexes, aging was associated with cardiac hypertrophy, LV concentric remodelling, and left atrial enlargement. Ejection fraction remained stable. Diastolic function was altered only in females.